# Blockchain Data Scalability and Retrieval Scheme Based on On-Chain Storage Medium for Internet of Things Data

Caoyi Yu [1,2], Niansong Mei [1,2,*], Chong Du [1,2] and Haotian Luo [1,2]

1   Shanghai Advanced Research Institute, Chinese Academy of Sciences, Shanghai 201210, China
2   University of Chinese Academy of Sciences, Beijing 100049, China
*   Correspondence: meins@sari.ac.cn; Tel.: +86-021-2032-5151

**Abstract:** The combination of blockchain and internet of things (IoT) technology realizes reliable storage of IoT data. However, the data stored on the blockchain (on-chain) face the problem of poor scalability and inefficient retrieval. In this paper, the on-chain data scalability schemes based on transactions and smart contracts are first proposed. Subsequently, on the basis of the above on-chain data scalability scheme based on transactions, an on-chain data index based on skip lists is proposed to improve the retrieval efficiency. The experimental results show that both the on-chain data scalability schemes achieve on-chain data scalability while reducing storage overhead. Meanwhile, the on-chain data index based on skip lists has significantly improved dynamic range retrieval efficiency and reduced the time complexity of single data retrieval to O(log(n)).

**Keywords:** blockchain; storage medium; on-chain data scalability; on-chain data retrieval; skip list index; internet of things

## 1. Introduction

Blockchain has the characteristics of decentralization, non-tampering and transparency. Compared to centralized systems, systems built on blockchain realize reliable storage of data and improve the security of the whole system [1]. Therefore, blockchain technology has attracted widespread attention from industry and academia, especially when combined with IoT technology [2]. The combination of blockchain and the IoT can be applied in more fields, such as IoT management [3], smart grids [4] and smart cities [5]. However, given the generation frequency and timeliness of IoT data, IoT data are extended and retrieved frequently, which imposes high requirements on the blockchain to store IoT data [6]. Meanwhile, the growing number of IoT devices generate huge amounts of data; for example, a typical smart city housing a population of 1 million is expected to generate 200 million GB data per day [7].

Such a huge amount of IoT devices and data rely on the blockchain to achieve reliable storage, making the blockchain encounter the challenge of inefficient on-chain data scalability and retrieval. However, data on the blockchain cannot be tampered with, which is the main reason why on-chain data are hard to extend [8]. Each node currently participating in the blockchain needs to back up all the data on the blockchain. When users update or add data, they need to upload all the data to the blockchain again, so duplicate data accumulate on the blockchain. Firstly, data redundancy caused by the inability to extend on-chain data wastes a lot of storage space. Secondly, the increasingly high storage cost increases the barrier to participation in blockchain projects, making fewer and fewer nodes in the blockchain. Therefore, lack of on-chain data scalability not only makes the blockchain defeat the original purpose of decentralization but also reduces the performance of the whole blockchain system [9]. Furthermore, although the blockchain can be used as a data storage layer, most current blockchain projects pursue efficient writing speeds and thus choose LevelDB based on LSM-tree. The LSM-tree structure has efficient write performance,

while its random read performance is not as impressive [10]. In addition, since LevelDB is a key-value database, there are not many query types that can be supported. However, both data reading performance and the number of query types affect the retrieval performance of on-chain data.

On-chain data scalability is currently implemented mainly by editing the on-chain data or deleting them and resubmitting them to the blockchain. However, all of the above methods require modifying the blockchain structure and verifying the operation, which is fatal for already deployed blockchain projects because of the high cost of rebuilding the project. Meanwhile, there are two types of solutions to improve the efficiency of on-chain data retrieval: one is based on external third parties and another is based on the blockchain itself. The former requires verification of the retrieved data.

To solve the above problems, this paper studies and analyzes the on-chain storage media for the first time. The on-chain data are eventually stored in the local database in the form of key-value pairs. However, on the blockchain, data are represented in a different form, such as transaction or smart contract, which are called on-chain storage media. Through studying on-chain storage media, this paper proposes two on-chain data scalability schemes, based on transaction and based on smart contract, respectively. Compared with existing schemes, the two on-chain data scalability schemes proposed in this paper achieve on-chain data scalability. They can also avoid massive modifications to the blockchain structure and implement direct compatibility with deployed blockchain projects. Moreover, in order to improve the retrieval efficiency of on-chain data, this paper proposes an on-chain data index based on skip lists on the basis of the above on-chain data scalability scheme based on transactions. By integrating the skip list index with blockchain, the new index proposed can implement range search and improve the retrieval efficiency of on-chain data. The main contributions of this paper are as follows:

- The on-chain storage media are studied and analyzed for the first time. According to the two different on-chain storage media, the on-chain data scalability schemes based on transactions and smart contracts are proposed to improve on-chain data scalability.
- On the basis of the above on-chain data scalability scheme based on transaction, an on-chain data index based on skip lists is proposed to implement range searches and improve the retrieval efficiency of on-chain data.
- The above two on-chain data scalability schemes and on-chain data index based on skip lists are fully experimented and evaluated on Ganache platform. The experimental results show that both on-chain data scalability schemes achieve on-chain data scalability and reduce storage overhead. Meanwhile, the scheme based on transactions is more efficient than the scheme based on smart contracts in terms of data retrieval; however, the scheme based on smart contracts is better in terms of functionality. Moreover, the on-chain data index based on skip lists has significantly improved dynamic range retrieval efficiency and has reduced the time complexity of single data retrieval to O(log(n)) compared with sequential retrieval and the B+ Tree index.

The paper is organized as follows: Section 2 introduces related works and preliminary concepts. Section 3 provides details of the on-chain storage media analysis and two on-chain data scalability schemes. Section 4 provides design details of the on-chain data index based on skip lists. Section 5 provides experimental verification of the above contents. Section 6 describes the conclusion and future work.

## 2. Related Works and Preliminary Concepts

### 2.1. On-Chain Data Scalability and Retrieval

The on-chain data scalability is currently implemented mainly by editing the on-chain data or deleting it and resubmitting it to the blockchain. Li, X.C. et al. proposed a redactable blockchain scheme based on unforgeable signatures, which can quickly confirm whether on-chain data has been illegally edited [11]. Aslam, S. et al. presented a RESTful decentralized storage framework that combines blockchain and distributed hash table (DHT) to enable on-chain data editing [12]. Pan, Y.Y. et al. introduced bilinear pairing into

Chameleon hash and designed a new generation method of Merkle tree based on it for data editing and validation [13]. Tang, Y.L. et al. proposed a redactable blockchain trust scheme based on reputation consensus and a one-way trapdoor function, which used SM2 asymmetric cryptography algorithm as the one-way trapdoor function to construct a new Merkle tree structure, ensuring the legality of data edits and deletions [14]. Feng. H.W. et al. proposed a collaborative replica deletion algorithm with greediness and established a replica deletion model based on the replica deletion loss and load state [15]. DMBlockChain is a deletable and modifiable blockchain scheme based on RVTrees and the multisignature scheme, which implements block data modification and deletion functions [16]. However, all of the above methods require modifications to the blockchain structure, which is fatal for already deployed blockchain projects. Meanwhile, they also require verification of delete and edit operations to guarantee the legitimacy of these operations, which needs expensive computational overhead. Improving on-chain data scalability through on-chain storage media can effectively avoid blockchain data redundancy. Compared with the above-mentioned schemes, the scheme based on on-chain storage media can avoid modifying the internal structure of the blockchain while being directly applicable to deployed blockchain projects.

There are two types of solutions to improve the efficiency of on-chain data retrieval: one is based on external third parties and another is based on the blockchain itself. The former exports blockchain data to external databases or cloud service platforms. VQL provides both efficient and verifiable data query services for blockchain by a cloud-based middle layer, which extracts on-chain data and efficiently reorganizes them in external databases [17]. Rahman, M.S. et al. proposed a blockchain-based framework which divides IoT data into on-chain and off-chain data for providing privacy preserving and verifiable query services to users [18]. The schemes based on external third parties can provide efficient retrieval services and rich retrieval methods. However, since third-party cloud services cannot be trusted, it is necessary to verify the data after retrieval.

The latter realizes efficient retrieval of on-chain data by adding indexes. Due to the fact that the data are obtained directly from the blockchain, it is unnecessary to validate the data. Xu, C. et al. proposed an authentication data structure based on accumulators, which supported dynamic aggregation of arbitrary query attributes and developed two new indexes to aggregate intra block and inter block data records to achieve efficient query verification [19]. Yan, D.K. et al. designed a dual-index based on the B+ Tree and the key-value pair inside blockchain through smart contract to support multiple query operations and improve query efficiency [20]. Wan, L. constructed the index directory of the BKV (B-key-value) tree storage structure by modifying the storage of the B-tree and designed the blockchain structure and query algorithm based on the BKV index directory to improve the efficiency of the electrical transaction query [21]. SEBDB predefines multiple attributes for each transaction in the blockchain and constructs a B+ Tree index for each attribute to query the transaction based on the attribute value [22]. EBTree is similar to SEBDB, but the difference is that EBTree constructs a B+ Tree index for all blocks in the blockchain [23]. Most of the above schemes improve the efficiency of on-chain data retrieval by adding B+ Tree indexes.

### 2.2. Ethereum and Smart Contract

In the blockchain 1.0 era, decentralized currency represented by Bitcoin was the main application. Ethereum is the most dominant distributed application platform after the development of blockchain technology to 2.0. Programs that run on Ethereum are called smart contracts, which were first proposed by Nick Szabo. Smart contracts are collections of code and data designed to execute transactions without relying on trusted third parties. Since operations are public and reliable on the blockchain, all transactions are traceable and irreversible. Ethereum is the first blockchain implementation to have a Turing complete virtual machine built into it, which implies that smart contracts in Ethereum can be used to perform any computational task [24]. Smart contracts have their own address just like

ordinary accounts, so it is logically consistent to call a smart contract and send transactions to other accounts [25].

### 2.3. Skip List

A skip list can be seen as a modified linear list. As opposed to a linear list that has only one pointer field for a node, a node in a skip list can have multiple pointer fields [26]. The structure of the skip list is shown in Figure 1. When a new node is inserted during the construction of a skip list, that node is inserted first at the bottom level. Then, it is promoted to the upper layer with one-half probability by the randomization algorithm and leaves a copy of the node in the lower layer until the randomization algorithm ends. When data retrieval is performed in the skip list, the higher-level nodes are retrieved first. If the higher-level retrieval fails, the lower-level is retrieved until the node is found [27]. Compared with the traditional index structure B+ Tree, a skip list does not require frequent and complex rebalancing operations when inserting and deleting data [28]. At the same time, skip list operations are concise and more suitable for parallel processing. In blockchain, on-chain data are placed linearly in the form of blocks, and therefore the structure of a skip list index naturally suits blockchain.

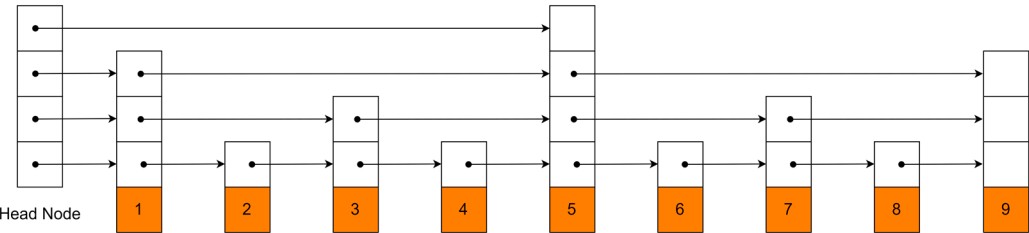

**Figure 1.** Skip list index structure.

## 3. Storage Medium and Scalability Scheme

What can be used as storage media in existing blockchains are transactions and smart contracts. In this section, the characteristics of transactions and smart contracts as storage media are discussed in detail, and two on-chain data scalability schemes are proposed.

### 3.1. Transaction

The transaction is the most basic component unit in the blockchain, where every action that occurs in the blockchain is stored. Multiple transactions form a block which is broadcast on the blockchain after being verified by the consensus algorithm. Therefore, using transactions as storage media to hold data is the most intuitive solution for holding data in the blockchain. However, a single transaction cannot hold large-scale data due to storage limitations, so the data must be split and stored in multiple transactions. According to the characteristics of the transaction, an on-chain data scalability scheme based on transactions is proposed.

If the split data are stored directly in the blockchain through transactions, the blocks must be scanned one by one to retrieve the data, resulting in poor retrieval efficiency, and the data already stored in the blockchain cannot be extended. As shown in Figure 2, the on-chain data scalability scheme based on transaction divides the content of the transaction into three parts: one part contains the hash value of the previous transaction, which is ParentHash; one part contains the group number of the data, which is GroupID; and the last part holds the data. When saving data, the ParentHash field of the first transaction is filled with all zeros, and the hash of the last transaction is recorded for subsequent data extensions. The GroupID field is the timestamp when the data was saved or expanded and is used as the group number for that group of data. When retrieving data, the hash of a transaction is requested and the data, GroupID and ParentHash are obtained from that hash. Retrieving is continued according to the ParentHash until the ParentHash obtained is all zeros. To restore the split data, all the obtained data are merged according to the

GroupID and the order of acquisition. Finally, when extending the data, the previously recorded hash is used as the content of the ParentHash field to create the transaction. At the end of the data extension, the hash value of the last transaction is updated.

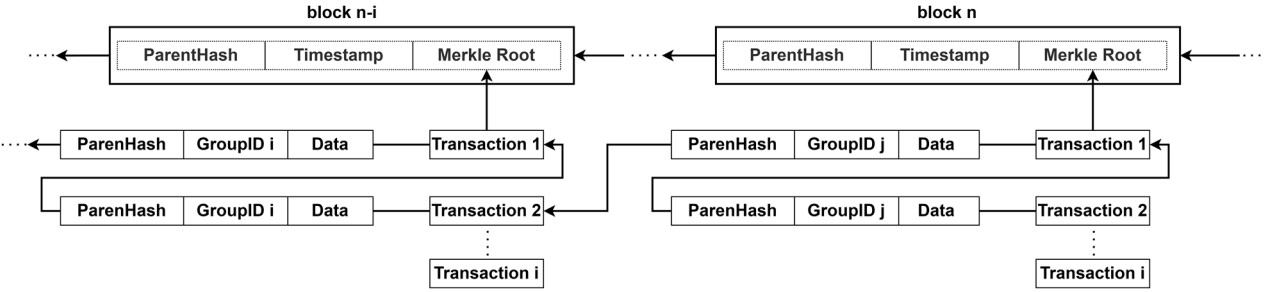

**Figure 2.** On-chain data scalability scheme based on transaction.

### 3.2. Smart Contract

Although it is obvious that the transaction can act as a storage medium, all things on the blockchain are eventually stored in LevelDB as key-value pairs, so the smart contract can also act as a storage medium in the blockchain. Since smart contracts are programs that run on the blockchain, they can provide a wealth of functionalities. After smart contracts are deployed to Ethereum, they are saved as accounts in the local LevelDB. Above all, the account address corresponding to the contract can be calculated in advance before the smart contract is deployed on Ethereum. Compared to transactions, smart contracts can both execute programs to provide various functions and act as accounts to receive transactions. However, it should be noted that smart contracts can only upload a limited amount of data to the blockchain due to the limitation of the Ethereum virtual machine.

According to the characteristics of the smart contract, an on-chain data scalability scheme based on smart contract is proposed. As shown in Figure 3, the main part of the data is stored in the smart contract. The contract address, which is generated after the contract is deployed on the blockchain, is used to extend the on-chain data. On-chain data scalability is implemented by storing extended data through transactions and connecting transactions to previously generated contract addresses. In this scheme, the transaction is divided into three parts, namely GroupID, Index and Extended data. The GroupID field and Extended data field are the same as for the on-chain data scalability scheme based on transaction. The reason for setting up the Index field is that the data are sliced before being saved, so the order of the data must be ensured when restored. Smart contracts are both an on-chain storage medium and a program that runs on the blockchain. Therefore, the smart contract in this scheme has the following main components: a function to obtain the data subject and a function to obtain the keywords or summary of the data. As the smart contract is code, the rest of the components are also represented in the smart contract in the form of functions, such as the function to encrypt the data.

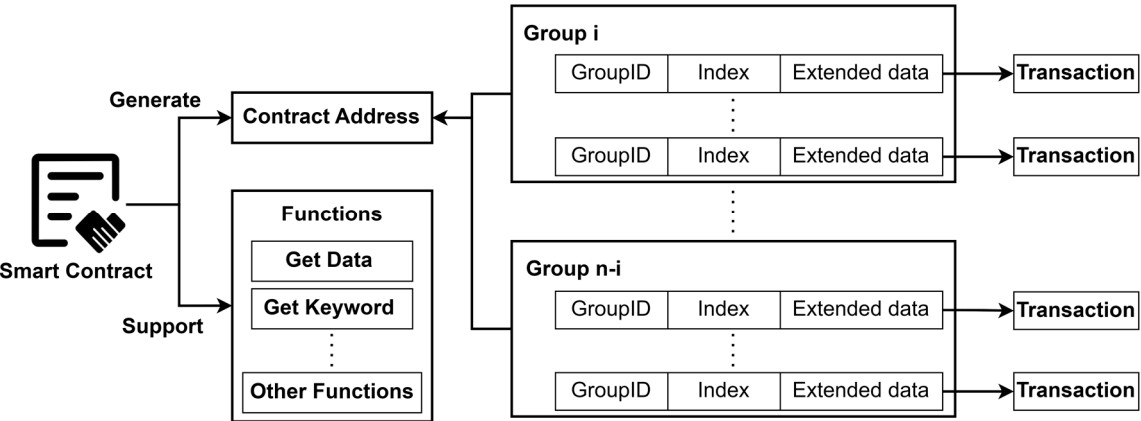

**Figure 3.** On-chain data scalability scheme based on smart contracts.

## 4. On-Chain Data Index Based on Skip Lists

IoT data related to smart cities uploaded to the blockchain need to be acquired and analyzed in real time to provide convenient services to citizens. The two on-chain data scalability schemes in the previous section enable on-chain data extension and reduce data redundancy on the blockchain. However, on-chain data can only be retrieved by traversing the entire blockchain. Consequently, on-chain data retrieval is inefficient. Therefore, in order to improve the retrieval efficiency of on-chain data, this paper proposes an on-chain data index based on skip lists on the basis of the above on-chain data scalability scheme based on transaction.

The on-chain data scalability scheme based on transaction divides the GroupID field in the transaction and identifies the group number of the data by the GroupID field. The GroupID field holds the timestamp of when the data were uploaded, so it is also possible to sort the data according to the time of upload in the GroupID field. As shown in Figure 4, on the basis of the above on-chain data scalability scheme based on transactions, the skip list structure is introduced. Compared to tree index structure such as B+ Tree, the skip list index, which is a linear list structure, does not require complex and heavy rebalancing operations when adding and removing nodes. Furthermore, the skip list index provides efficient range searches in addition to efficient single data retrieval efficiency. The GroupID field in the transaction is used as a keyword to create the node in the skip list index. Through constructing a skip list index on the blockchain and using the GroupID field as the keywords for on-chain data retrieval, the range search of on-chain data is realized and the retrieval efficiency of on-chain data is improved.

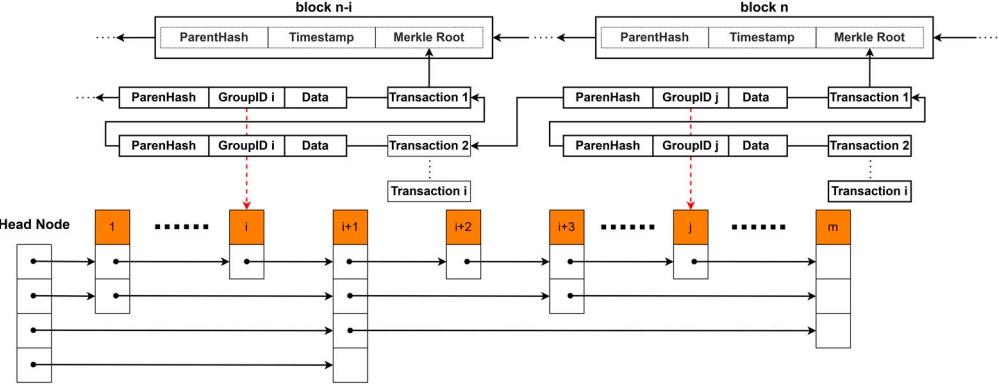

**Figure 4.** On-chain data index based on skip list.

## 5. Experiment

In this section, the above two on-chain data scalability schemes and on-chain data index based on skip lists are fully experimented and evaluated on the Ganache platform

### 5.1. Experimental Setup

All experiments were conducted on the same server with an Intel Core i7-10710U and 16 GB RAM and running Ubuntu 20.04 64-bit operating system. The blockchain system is Ganache-CLI v6.12.2, an Ethernet emulation network with ten nodes by default. The smart contract code is written in Solidity 0.8.11, and the index and system are written and built in Java. Web3j 4.8.7 was used as the middleware for the interaction between the system and the blockchain. In this section, the scalability overhead and data retrieval performance of on-chain data scalability schemes based on transactions and smart contracts are first tested, respectively, and the differences between the two schemes are analyzed. Then, the retrieval efficiency of on-chain data index based on skip lists, including the efficiency of single data searches and dynamic range searches, is then tested and compared with sequential retrieval and B+ tree index.

### 5.2. Performance of Two On-Chain Data Scalability Schemes

The extended data size set in the experiments is gradually increased from 256 Bytes to 2048 Bytes, and the initial data size in the experiments is 512 Bytes. The scalability overheads of the two schemes are tested first. Figure 5 presents the execution times of the two scalability schemes in this paper for on-chain data extension. According to the results in Figure 5, as the extended data increases, so does the execution time; however, the execution efficiency is essentially the same for the two scalability schemes. Compared to the existing schemes [11–14], the two scalability schemes are more efficient to extend because they do not require verification of deleting and editing operations. Figure 6 illustrates the result of the smart contract deployment. The storage overheads of the two scalability schemes in this paper are then tested. A transaction is set to store 128 bytes of data in it. Table 1 presents the number of transactions created when the on-chain data are extended. The scalability scheme based on smart contract creates only one transaction when storing the initial data because the data subject is stored within the smart contract code. According to the data in Table 1, the two on-chain data scalability schemes in this paper reduce the storage overhead significantly compared to the on-chain data scalability of traditional blockchain. Therefore, the on-chain data scalability schemes based on transactions and based on smart contracts effectively improve the scalability of the current blockchain.

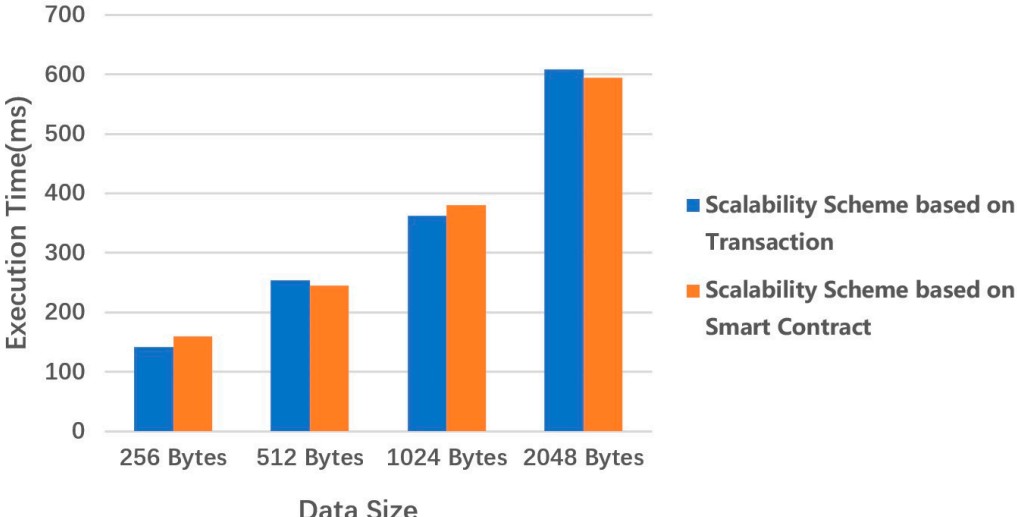

**Figure 5.** Execution times of the two scalability schemes.

```
eth_sendTransaction

 Transaction: 0xa5331c6c0eff6bf207979fd616bf74a9898f45892910a58badf39c18b6d548b4
 Contract created: 0x83031d01f34fa564a48f1f33d8a9d23d88075e0c
 Gas usage: 346797
 Block Number: 373
 Block Time: Wed Mar 01 2023 01:38:50 GMT+0800

eth_getTransactionReceipt
```

**Figure 6.** Result of the smart contract deployment.

**Table 1.** Transaction usage.

| Scheme | Initial Data | 256 Bytes | 512 Bytes | 1024 Bytes | 2048 Bytes |
|---|---|---|---|---|---|
| Transaction Scheme | 4 | 6 | 10 | 18 | 34 |
| Smart Contract Scheme | 1 | 3 | 7 | 15 | 31 |
| Traditional Scheme | 4 | 10 | 20 | 38 | 72 |

Then the single-point query efficiency and multi-point query efficiency are tested separately. The multi-point query efficiency results from 100 threads executing simultaneously. Figure 7 illustrates the execution time of the on-chain data scalability scheme based on transactions. With the increase in extended data, the execution times of both single-point and multi-point query increase; however, the increase of single-point query time is not significant compared with multi-point query. Figure 8 demonstrates the execution time of the on-chain data scalability scheme based on smart contracts. The result is similar to the transaction-based scheme, with execution time increasing as the extended data increase. The results in Figures 7 and 8 also indicate that on-chain data scalability based on transactions and based on smart contracts is feasible.

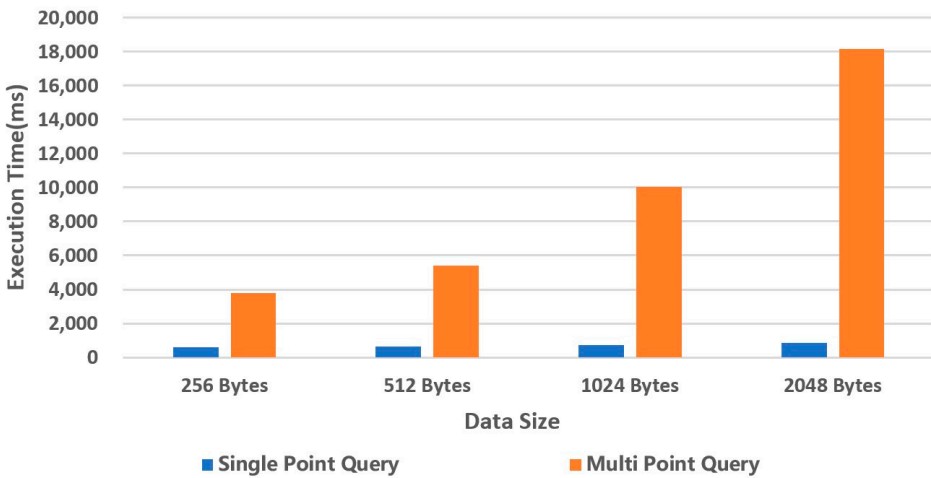

**Figure 7.** Execution time of the on-chain data scalability scheme based on transaction.

Figure 9 illustrates the comparison of the single-point query execution time of the two schemes. It is obvious from Figure 9 that the transaction-based scheme is significantly more efficient than the smart-contract-based scheme. After analysis, the on-chain data scalability scheme based on transaction only needs to retrieve the raw data and the extended data directly from LevelDB. However, for the smart-contract-based scalability scheme, since the raw data are stored in the smart contracts, obtaining the raw data requires reading the smart contract code from LevelDB to the Ethereum virtual machine and then running the function to obtain it, which increases the execution time. Nevertheless, the on-chain data scalability scheme based on smart contracts is significantly better than the transaction-based scalability scheme in terms of functionality because of its ability to execute functions.

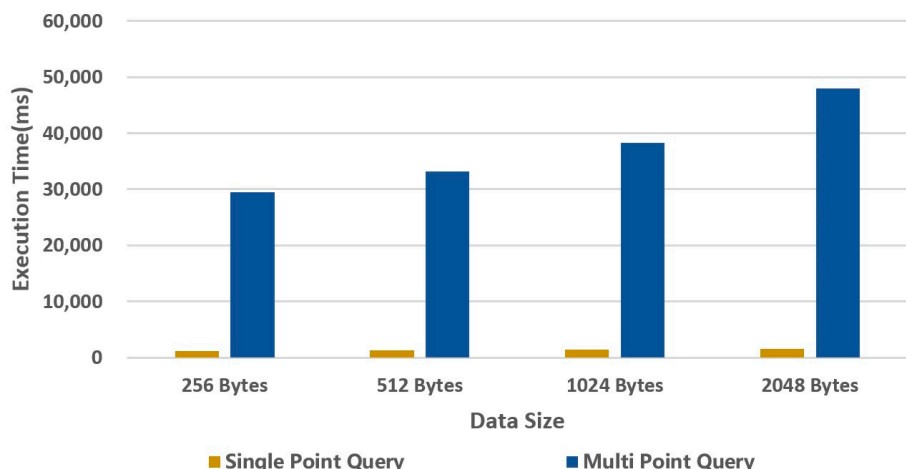

**Figure 8.** Execution time of the on-chain data scalability scheme based on smart contract.

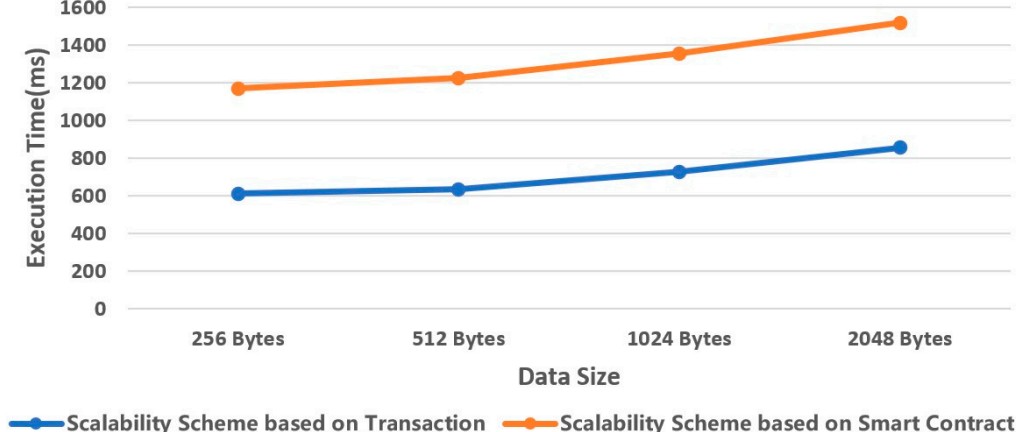

**Figure 9.** Comparison of the execution time of the two schemes.

*5.3. Performance of On-Chain Data Index Based on Skip Lists*

This experiment tests the retrieval efficiency of on-chain data index based on skip lists by gradually increasing the number of blocks and compares it with the efficiency of sequential retrieval and B+ Tree index. The retrieval efficiency of single data is first compared and the results are illustrated in Figure 10a,b. In single data retrieval, the on-chain data index based on skip lists is obviously more efficient than sequential retrieval and B+ Tree index. The time complexity of sequential retrieval is O(n), while the time complexity of the skip list and B+ Tree are both O(log(n)). It can also be observed from Figure 10 that the execution time decreases as the number of blocks increases. This is due to the fact that the skip list index is constructed through a random function. The more data are available, the higher the probability of obtaining the high-level number and the more efficient the data retrieval is.

Figure 11 shows the retrieval efficiency of the dynamic range, where multiple sets of data are set, and the number of blocks occupied is 100 and 1000. From Figure 11, it is observed that the execution time of sequential retrieval remains the same as the retrieval range increases, while the execution time of the skip list index gradually increases. Until the range reaches 100%, the execution time of skip list index and sequential retrieval are basically the same when there are more blocks. The experimental results in Figures 10 and 11 demonstrate that the retrieval efficiency of on-chain data can be significantly improved by constructing a skip list index.

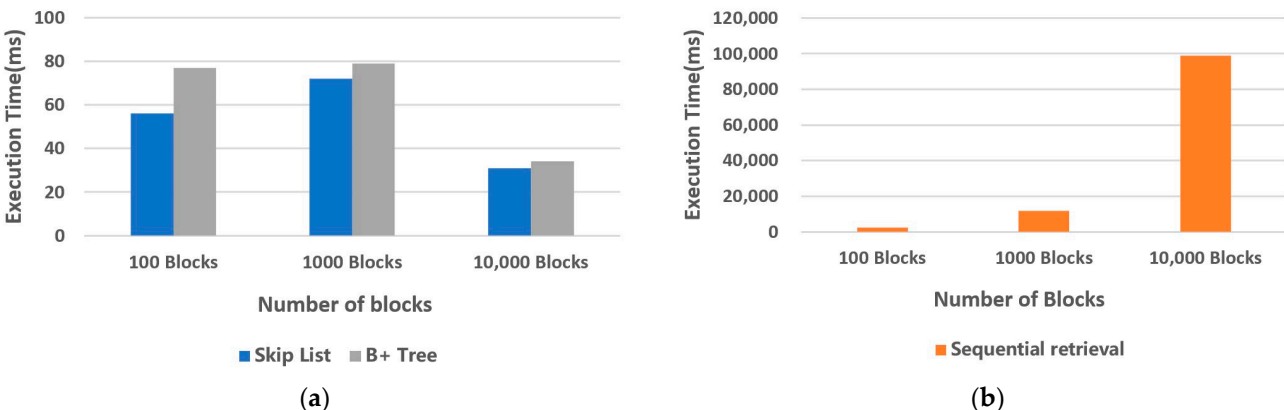

**Figure 10.** Retrieval efficiency of single data. (**a**) Index retrieval efficiency; (**b**) Sequential retrieval efficiency.

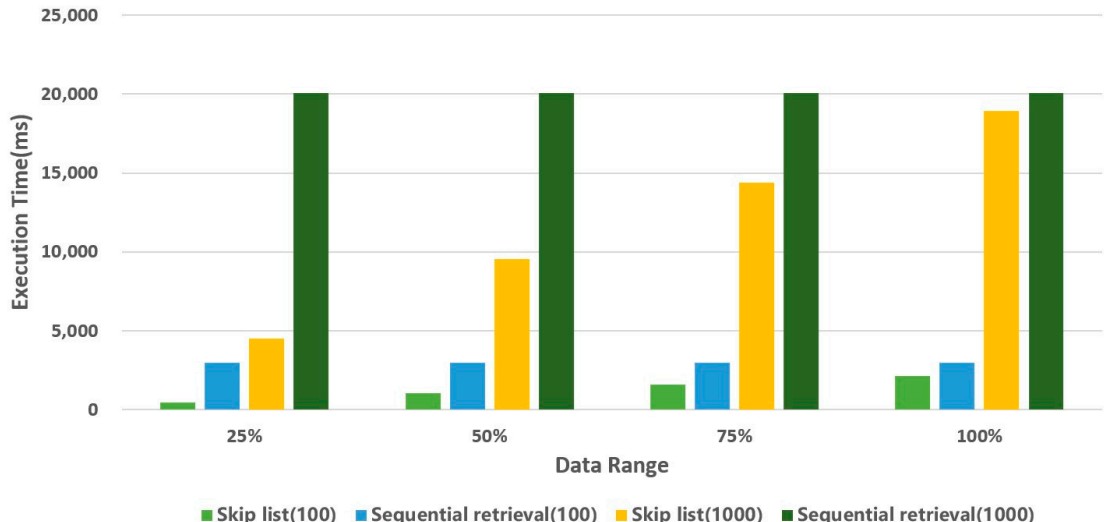

**Figure 11.** Retrieval efficiency of the dynamic range.

## 6. Conclusions

Currently, on-chain data scalability and retrieval are attracting a lot of attention from academia and industry. In this paper, two on-chain storage media, transactions and smart contracts, are first studied and analyzed, and two on-chain data scalability schemes are proposed to improve on-chain data scalability. Moreover, on the basis of the above on-chain data scalability scheme based on transactions, an on-chain data index based on skip lists is proposed to realize the range search of on-chain data and improve the retrieval efficiency of on-chain data. In the future, we will work on optimizing two on-chain data scalability schemes, especially the smart contract-based scheme, to improve the efficiency of on-chain data retrieval.

**Author Contributions:** Conceptualization, C.Y., N.M. and C.D.; methodology, C.Y. and N.M.; software, C.Y.; validation, C.Y. and N.M.; formal analysis, C.Y.; investigation, C.Y. and N.M.; resources, C.D.; data curation, C.Y. and H.L.; writing—original draft preparation, C.Y.; writing—review and editing, C.Y.; visualization, C.Y.; supervision, C.Y.; project administration, C.Y. All authors have read and agreed to the published version of the manuscript.

**Funding:** This research received no external funding.

**Data Availability Statement:** The data presented in this study are available on request from the corresponding author.

**Conflicts of Interest:** The authors declare no conflict of interest.

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
