# Peer review of "Blockchain Data Scalability and Retrieval Scheme Based on On-Chain Storage Medium for Internet of Things Data"

_electronics, doi:10.3390/electronics12061454_

Round 1

Reviewer 1 Report

Few observations below:

1. Title must be justified in the Abstract

2. Keywords are very poor

3. Reduce the length of the Abstract

4. Avoid description in the Abstract

5. Is it only applicable to smart cities?

6. What may happen if you remove the term "Smart City"?

7. Define components of smart contracts.

8. Redraw all the figures in experiments

9. split fig 8 in two

10. What may happen by increasing blocks?

11. in fig 9, it is not agreed with100% data range

12. The conception must be clear

Reviewer 2 Report

- The paper focuses on analyzing the on-chain storage medium for scalability and retrieval.

- For improving the on-chain data scalability, the authors propose 2 techniques: one based on transactions and the other based on smart contracts.

The results indicate that both the proposed schemes support data scalability. The approach based on the transaction is more efficient for data retrieval than the scheme based on smart contract.

- Good visualization 

- Paper is well structured and well written. 

- A direct comparison with the past work done can be useful. 

Reviewer 3 Report

The authors provide the on-chain storage medium is studied and analyzed for the first time. For two different storage media, the on-chain data scalability schemes based on transaction and based on smart contract respectively are proposed to improve on-chain data scalability. This a very nice and easy-to-read paper on a topic very suitable for this journal. My main concerts are listed below:

1- I enclose a pdf with different comments to be solved.

2- More technical papers about IoT and blockchain

[1] A Decentralized Location-Based Reputation Management System in the IoT Using Blockchain. IEEE Internet of Things Journal, 9(16), 15100-15115.

[2] Internet of things. Manual of digital earth, 387-423.
